# Ubiquitination-dependent control of sexual differentiation in fission yeast

**Fabrizio Simonetti[1,2], Tito Candelli[1,2], Sebastien Leon[3], Domenico Libri[1], Mathieu Rougemaille[1†]\***

[1]Institut Jacques Monod, Team "Metabolism and Function of RNA in the Nucleus", CNRS, UMR7592, Université Paris-Diderot, Sorbonne Paris Cité, Paris, France; [2]Université Paris-Saclay, Gif-sur-Yvette, France; [3]Institut Jacques Monod, Team "Membrane Trafficking, Ubiquitin and Signaling", CNRS, UMR9198, Université Paris-Diderot, Sorbonne Paris Cité, Paris, France

**Abstract** In fission yeast, meiosis-specific transcripts are selectively eliminated during vegetative growth by the combined action of the YTH-family RNA-binding protein Mmi1 and the nuclear exosome. Upon nutritional starvation, the master regulator of meiosis Mei2 inactivates Mmi1, thereby allowing expression of the meiotic program. Here, we show that the E3 ubiquitin ligase subunit Not4/Mot2 of the evolutionarily conserved Ccr4-Not complex, which associates with Mmi1, promotes suppression of meiotic transcripts expression in mitotic cells. Our analyses suggest that Mot2 directs ubiquitination of Mei2 to preserve the activity of Mmi1 during vegetative growth. Importantly, Mot2 is not involved in the constitutive pathway of Mei2 turnover, but rather plays a regulatory role to limit its accumulation or inhibit its function. We propose that Mmi1 recruits the Ccr4-Not complex to counteract its own inhibitor Mei2, thereby locking the system in a stable state that ensures the repression of the meiotic program by Mmi1.

DOI: https://doi.org/10.7554/eLife.28046.001

**\*For correspondence:**
mathieu.rougemaille@i2bc.paris-saclay.fr

**Present address:** [†]Institute for Integrative Biology of the Cell (I2BC), CEA, CNRS, Univ. Paris-Sud, Université Paris-Saclay, Gif-sur-Yvette, France

**Competing interests:** The authors declare that no competing interests exist.

## Introduction

The cell cycle switch from mitosis to meiosis is associated with profound changes in gene expression. In yeast, initiation of meiosis occurs upon nutrient starvation and depends on well-characterized signalling pathways (*Yamamoto, 2010*). Several hundred genes are induced thanks to specific transcription factors, which define the key steps of the meiotic program (*Mata et al., 2007*). Previous work in *S. pombe* has revealed the existence of an additional mechanism that controls the onset of meiosis. An RNA degradation system selectively eliminates meiosis-specific transcripts produced during the mitotic cell cycle, thereby inhibiting sexual differentiation (*Harigaya et al., 2006*). Essential to this regulatory process is Mmi1 (meiotic mRNA interceptor 1), a member of the conserved YTH family of RNA-binding proteins that localizes exclusively to the nucleus. Mmi1 recognizes a *cis*-acting region within targeted mRNAs, called DSR (Determinant of Selective Removal), which confers nuclear exosome-mediated degradation. DSR regions are enriched in repeats of the hexanucleotide motif UNAAAC to which Mmi1 binds to via its C-terminal YTH domain (*Yamashita et al., 2012*; *Wang et al., 2016*; *Chatterjee et al., 2016*; *Wu et al., 2017*; *Touat-Todeschini et al., 2017*). Mmi1 also associates to several coding and non-coding RNAs with fewer UNAAAC motifs, indicating flexibility in target recognition (*Kilchert et al., 2015*; *Touat-Todeschini et al., 2017*).

Several factors identified both by genetic screens and biochemical analyses, cooperate with Mmi1 to promote meiotic mRNA suppression during vegetative growth. Previous studies demonstrated a role for components of the 3'-end processing machinery as well as the canonical poly(A) polymerase Pla1 and the poly(A) binding protein Pab2 (*St-André et al., 2010*; *Yamanaka et al., 2010*). Mechanistically, it was first proposed that Mmi1 promotes hyperadenylation of targeted

transcripts and binding of Pab2 onto mRNAs poly(A) tails, which in turn recruits the nuclear exosome subunit Rrp6 for degradation. Mmi1 was next found to associate and cooperate with the multi-subunit MTREC complex to mediate meiotic mRNAs suppression (*Sugiyama and Sugioka-Sugiyama, 2011*; *Yamashita et al., 2012*; *Lee et al., 2013*; *Egan et al., 2014*; *Zhou et al., 2015*). Core components of this complex include the zinc finger-containing protein Red1 and the Mtr4-like RNA helicase Mtl1 (*Lee et al., 2013*; *Egan et al., 2014*; *Zhou et al., 2015*). MTREC is crucial for meiotic mRNA degradation and has been proposed to physically bridge RNA-bound Mmi1 to the nuclear exosome. Recent studies also reported an interaction between Mmi1 and the Ccr4-Not complex (*Cotobal et al., 2015*; *Ukleja et al., 2016*; *Sugiyama et al., 2016*; *Stowell et al., 2016*), the major cytoplasmic mRNA deadenylation machinery conserved from yeast to humans (*Wahle and Winkler, 2013*). However, although Mmi1 recruits Ccr4-Not to its RNA targets in vivo (*Cotobal et al., 2015*) and stimulates its deadenylation activity in vitro (*Stowell et al., 2016*), this function of the complex is not required for the turnover and the translation of DSR-containing meiotic mRNAs (*Cotobal et al., 2015*; *Sugiyama et al., 2016*).

A subset of Mmi1-regulated genes is covered by repressive chromatin marks (i.e. dimethylated histone H3 lysine 9 or H3K9me2) during vegetative growth, which disappear upon developmental or environmental cues (*Zofall et al., 2012*; *Hiriart et al., 2012*; *Tashiro et al., 2013*; *Shah et al., 2014*). Mmi1 and the MTREC subunit Red1 recruit the H3K9 methyltransferase Clr4 to mediate H3K9me2 deposition and direct components of the RNAi machinery to these loci (*Zofall et al., 2012*; *Hiriart et al., 2012*; *Tashiro et al., 2013*; *Shah et al., 2014*). The function of Mmi1, Clr4 and RNAi factors in suppressing the expression of meiotic mRNAs in vegetative cells also prevents the incidence of chromosome missegregation events (*Folco et al., 2017*). However, the contribution of heterochromatin and RNAi machineries to the silencing of Mmi1 targets is moderate. Rather, the post-transcriptional degradation pathway mediated by Mmi1, MTREC and the exosome is predominant (*Harigaya et al., 2006*; *Lee et al., 2013*; *Ard et al., 2014*).

Upon nutritional starvation, Mmi1 is sequestered in an RNP complex, which allows translation of meiotic mRNAs and progression of the cell through meiosis (*Harigaya et al., 2006*). This inhibitory complex includes the nucleocytoplasmic shuttling RNA-binding protein Mei2, required for pre-meiotic DNA synthesis and the first meiotic division, and the DSR-containing lncRNA meiRNA, encoded by the *sme2+* gene (*Watanabe and Yamamoto, 1994*; *Sato et al., 2001*; *Shimada et al., 2003*). Sequestration of Mmi1 by the Mei2-meiRNA complex occurs at the *sme2+* locus (*Shimada et al., 2003*; *Shichino et al., 2014*) and is essential for entry into meiosis and sexual differentiation (*Harigaya et al., 2006*; *Shichino et al., 2014*).

Mitotic cells exploit transcriptional and post-translational mechanisms to control Mei2 abundance and activity. *mei2+* expression depends on the meiosis-specific transcription factor Ste11 (*Sugimoto et al., 1991*) and its activity and stability are regulated by phosphorylation via two protein kinases, Pat1 (also known as Ran1) and Tor2, both of which are essential for vegetative growth and inhibition of sexual differentiation (*Kitamura et al., 2001*; *Alvarez and Moreno, 2006*; *Otsubo et al., 2014*). Phosphorylated forms of Mei2 are inhibited by 14-3-3 family proteins (*Sato et al., 2002*) and are targeted for degradation by the proteasome in an ubiquitination-dependent manner (*Kitamura et al., 2001*; *Otsubo et al., 2014*).

Despite recent progresses, a full understanding of how Mmi1 cooperates with its cofactors during vegetative growth to prevent initiation of the meiotic program is still elusive. Using affinity purification and co-immunoprecipitation assays, we show that Mmi1, but not MTREC, stably associates in vivo with the evolutionarily conserved Ccr4-Not complex. This interaction is functionally relevant because integrity of the Ccr4-Not complex is required for meiotic mRNA suppression during vegetative growth. Surprisingly, we show that the RNA deadenylases Ccr4 and Caf1/Pop2 are dispensable for controlling the levels of meiotic transcripts, while the E3 ubiquitin ligase subunit Not4/Mot2 is essential for this process. Importantly, biochemical and genetic analyses suggest that Mot2 ubiquitinates the Mmi1 inhibitor Mei2 to functionnaly inactivate the latter or to limit its steady state levels. Our results reveal that Mot2 is not responsible for the constitutive pathway of Mei2 turnover, which depends on the E3 ubiquitin ligase Ubr1, but negatively controls Mei2 to preserve the activity of Mmi1. We propose that Mmi1 recruits the Ccr4-Not complex to fine-tune the levels or the activity of its own inhibitor Mei2, thereby sustaining efficient meiotic mRNAs suppression. Our data unveil an important role for the conserved E3 ligase Not4/Mot2 in providing an additional level of control to the repression of the meiotic program in vegetative cells.

## Results

### The RNA-binding protein Mmi1, but not MTREC, associates with the Ccr4-Not complex in vivo

To obtain a comprehensive view of Mmi1 protein partners during vegetative growth, we affinity-purified a C-terminally TAP-tagged version of the protein and analyzed interacting proteins by mass spectrometry (*Figure 1A–B*, *Figure 1—figure supplement 1A*). We identified several factors that co-purify with Mmi1, even when extracts were treated with RNases (*Figure 1B*, *Figure 1—source data 1*). As previously reported (*Yamashita et al., 2012*; *Lee et al., 2013*; *Egan et al., 2014*; *Zhou et al., 2015*; *Sugiyama et al., 2016*), we found the Mmi1 cofactor Erh1, components of the MTREC complex as well as splicing factors. The most abundant interacting factors were the subunits of the Ccr4-Not complex (*Figure 1B*), which was also reported independently while this work was in progress (*Cotobal et al., 2015*; *Ukleja et al., 2016*; *Sugiyama et al., 2016*; *Stowell et al., 2016*). Other associated proteins include subunits of the proteasome, transcription and chromatin remodelling factors as well as components of signalling pathways. We further focused on the functional implications of the interaction between Mmi1 and Ccr4-Not.

The Ccr4-Not complex assembles around its scaffolding component Not1 and contains three catalytic subunits, the Ccr4 and Caf1/Pop2 deadenylases and the E3 ubiquitin ligase Not4/Mot2 (*Ukleja et al., 2016*). We validated the interactions revealed by the MS analysis using co-immunoprecipitation experiments. Mmi1 efficiently pulled down Not1, Caf1 and Mot2 in an RNA-independent manner (*Figure 1C*, *Figure 1—figure supplement 1B–C* ). These interactions were preserved in the absence of Red1, which is required for MTREC integrity, possibly supporting the existence of alternative Mmi1-containing complexes. Consistent with this, Mtl1 and Red1 did not interact with Not1 and Caf1 in our assays (*Figure 1D*, and *Figure 1—figure supplement 1D*). Together, our data support the notion that Mmi1 associates independently with MTREC and Ccr4-Not during vegetative growth.

### The E3 ubiquitin ligase Mot2 of the Ccr4-Not complex is required for meiotic mRNAs suppression

The Ccr4-Not complex localizes throughout the cell and is involved in essentially all steps in gene expression, including chromatin modification, transcription, nuclear and cytoplasmic mRNA degradation, translational repression, protein degradation and quality control (*Miller and Reese, 2012*; *Collart, 2016*). To assess whether Ccr4-Not plays a role in the suppression of meiotic mRNAs by Mmi1 during vegetative growth, we first examined steady state levels of meiotic transcripts by RT-qPCR in strains deleted for its non-essential subunits (i.e. all but the scaffolding component Not1). Neither the RNA deadenylases Ccr4 and Caf1 nor the E3 ubiquitin ligase Mot2 were required for meiotic mRNAs degradation when cells were grown in rich medium, in contrast to the nuclear exosome subunit Rrp6 (*Figure 2A*). However, a significant role for the complex became apparent when cells were cultured in minimal medium. Indeed, several selected meiotic mRNAs, including *mei4+*, *ssm4+* and *mcp5+* were upregulated specifically in the *mot2Δ* mutant (*Figure 2B*), in some cases to levels comparable to those observed in *rrp6Δ* cells (e.g. *mcp5+*). Interestingly, the lack of meiotic RNAs stabilization in *caf1Δ*, *ccr4Δ* and *caf1Δ ccr4Δ* mutants, indicates that the deadenylation activity of the complex is not required for RNA degradation (*Figure 2B*, data not shown), consistent with recent studies (*Cotobal et al., 2015*; *Sugiyama et al., 2016*). Rather, the specific requirement for the Mot2 subunit strongly suggests the unprecedented possibility that meiotic mRNA suppression involves protein ubiquitination.

Previous work showed that the RNA deadenylases Ccr4 and Caf1 associate with DSR-containing meiotic mRNAs (*Cotobal et al., 2015*). Using RNA-immunoprecipitation assays, we found that the E3 ubiquitin ligase Mot2 interacts with the *mei4+* and *mcp5+* transcripts to the same extent as the Caf1 subunit, albeit less efficiently than Mmi1 (*Figure 2—figure supplement 1*). This is consistent with the notion that Mmi1 recruits the whole Ccr4-Not complex to its RNA targets (*Stowell et al., 2016*).

To establish whether the requirement for Mot2 in meiotic mRNA suppression is general, we analyzed the transcriptomes of wild type, *mot2Δ* and *rrp6Δ* strains grown in minimal medium by RNA-seq. Differentially expressed genes were determined by taking into account the fold enrichment of

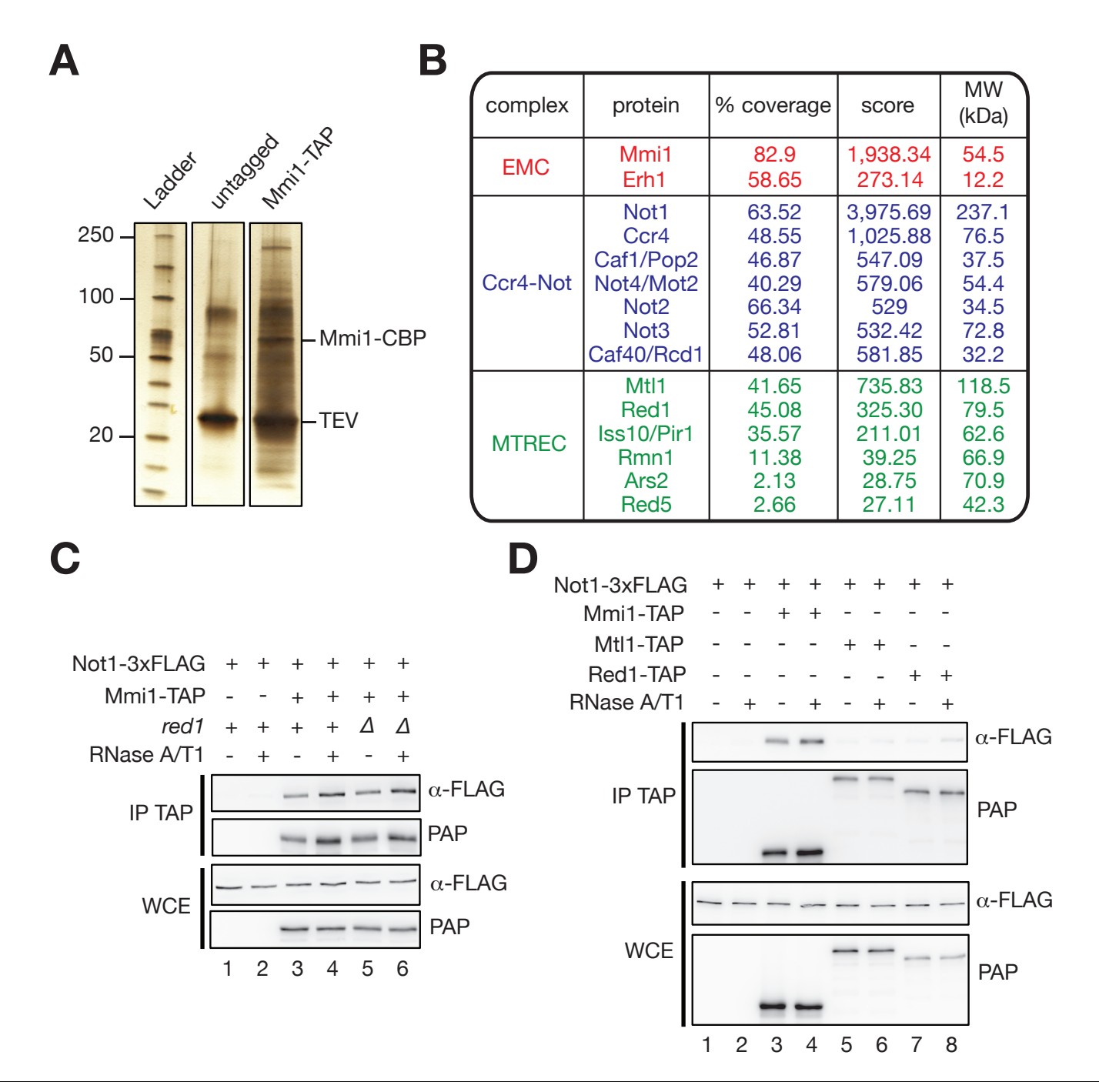

**Figure 1.** The RNA-binding protein Mmi1, but not MTREC, associates with the Ccr4-Not complex in vivo. (A) Silver-stained SDS polyacrylamide gel showing proteins co-eluting with TAP-tagged Mmi1 in minimal medium (EMM0.5X) after one-step affinity purification. Extracts were treated with RNaseA/T1 before immunoprecipitation and TEV cleavage. As a control, extracts from cells expressing untagged protein were used. The positions of the bait protein (Mmi1-CBP) and TEV are indicated. (B) Results of liquid chromatography-tandem mass spectrometry (LC-MS/MS) analysis of Mmi1-TAP associated proteins. The percentage of sequence coverage, scores (i.e. significance of the identified peptides represented as the -log10 of the Posterior Error Probability provided by the Percolator algorithm) and molecular weights are indicated. (C) Western blot showing that Not1-3xFLAG co-immunoprecipitates with Mmi1-TAP in minimal medium (EMM0.5X) in an RNA- and Red1-independent manner. (WCE) Whole Cell Extract; (IP) Immunoprecipitation. (D) Western blot showing that Not1-3xFLAG co-immunoprecipitates with Mmi1-TAP in minimal medium (EMM0.5X), but not with Mtl1-TAP and Red1-TAP. (WCE) Whole Cell Extract; (IP) Immunoprecipitation.

DOI: https://doi.org/10.7554/eLife.28046.002

*Figure 1 continued on next page*

*Figure 1 continued*

The following source data and figure supplement are available for figure 1:

**Source data 1.** Complete lists of Mmi1 protein partners in wt and *mot2Δ* cells.

DOI: https://doi.org/10.7554/eLife.28046.004

**Figure supplement 1.** The RNA-binding protein Mmi1, but not MTREC, associates with the Ccr4-Not complex in vivo.

DOI: https://doi.org/10.7554/eLife.28046.003

transcripts in mutant versus wild type cells (<or > 1.5 fold) and their statistical significance (p-value<5E-2) (*Figure 2C*). This analysis revealed increased expression for 622 and 393 transcripts in the *mot2Δ* and *rrp6Δ* mutants respectively (*Figure 2—source data 1*). A highly significant overlap of 100 transcripts indicates that Rrp6 and Mot2 function to repress expression of a common set of targets (*Figure 2D*). Importantly, a large fraction of these RNAs contain DSR elements and are known targets of Mmi1, MTREC and the nuclear exosome during vegetative growth (*Figure 2C–D*) (*Lee et al., 2013*; *Hiriart et al., 2012*). Gene ontology analysis indicated that upregulated transcripts in *mot2Δ* cells fall into three distinct functional categories (*Figure 2—source data 1*). The most significant group included genes involved in meiosis or that show an increase in expression upon meiotic induction. The second group includes genes required for mating and conjugation, and the third class contains genes encoding protein kinases (*Figure 2—source data 1*). However, these two latter categories were not over-represented in the *rrp6Δ* mutant, suggesting that the genes they belong to are not regulated in an Mmi1-dependent manner. Whether this relates to other functions of Mot2 and/or indirect effects remains to be determined. Together, our results reveal a global requirement for Mot2 in suppressing meiotic transcripts, among which Mmi1 targets are significantly enriched.

## The E3 ubiquitin ligase Mot2 negatively affects the levels of the Mmi1 inhibitor Mei2

The E3 ubiquitin ligase Not4/Mot2 has various functions in protein metabolism. Previous work showed that Mot2 controls protein turnover through ubiquitination and proteasome-dependent degradation to regulate chromatin modification, DNA replication, transcription as well as translation (*Panasenko et al., 2006*; *Laribee et al., 2007*; *Mersman et al., 2009*; *Haworth et al., 2010*; *Cooper et al., 2012*; *Sun et al., 2015*; *Laribee et al., 2015*; *Brönner et al., 2017*). Other studies revealed roles in the functional integrity of the proteasome (*Panasenko and Collart, 2011*) and in translational quality control (*Dimitrova et al., 2009*; *Halter et al., 2014*; *Preissler et al., 2015*). Mot2 also mediates non-destabilizing protein ubiquitination (*Panasenko and Collart, 2012*), highlighting its functional versatility.

The requirement of Mot2 for meiotic mRNAs suppression might reflect the need for inhibiting or degrading a negative regulator of nuclear RNA decay during vegetative growth. One possible candidate is the RNA-binding protein Mei2, which functions to inactivate Mmi1 (*Harigaya et al., 2006*). We therefore hypothesized that Mmi1 might recruit the Ccr4-Not complex to Mei2 to promote its ubiquitination by Mot2.

We first determined Mei2 protein levels in wild type and *mot2Δ* cells grown in different conditions. In the presence of Mot2, Mei2 was hardly detectable in rich medium and accumulated only to low levels in minimal medium (*Figure 3A*, lanes 2 and 5). Deletion of *mot2+* resulted in a slight accumulation of Mei2 in rich medium, reaching levels comparable to those observed in a wild type strain grown in minimal medium (*Figure 3A*, compare lanes 3 and 5). Importantly, Mei2 levels increased substantially in *mot2Δ* cells cultured in minimal medium (*Figure 3A*, lane 6), which correlates with the requirement for Mot2 in meiotic mRNAs suppression in these conditions (*Figure 2B*).

Higher Mei2 levels in *mot2Δ* cells might result from an effect on the production (or stability) of *mei2+* mRNAs. Indeed, we found increased *mei2+* transcript levels in the *mot2Δ* mutant (*Figure 3—figure supplement 1A*), possibly as a result of a direct or indirect negative control exerted by Mot2 on the promoter of the *mei2+* gene. To assess whether Ccr4-Not regulates Mei2 levels in a transcription-independent manner, we replaced the endogenous *mei2+* promoter with the *nmt1+* promoter, which is active in minimal medium lacking thiamine. In this context, we also detected a four-fold increase of the Mei2 protein in the *mot2Δ* mutant relative to wild type cells (*Figure 3B–C*), while *mei2+* mRNAs accumulated to only a limited extent (1.6 fold) (*Figure 3D*). Moreover, defective

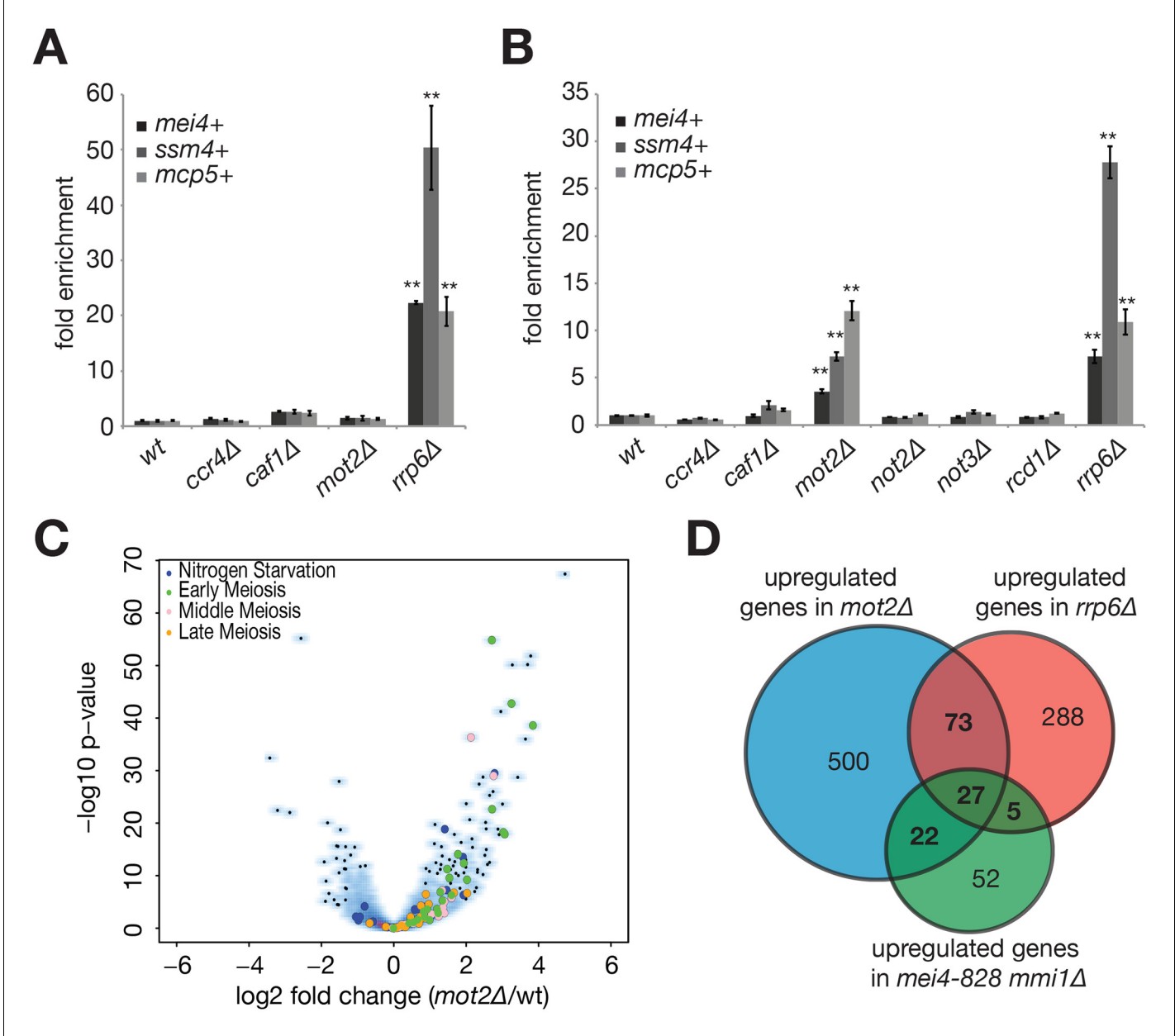

**Figure 2.** The E3 ubiquitin ligase Mot2 is required for meiotic mRNAs suppression. (**A**) RT-qPCR analysis of the *mei4+*, *ssm4+* and *mcp5+* meiotic transcripts in cells grown in rich medium (YE) and deleted for Ccr4, Caf1, Mot2 or Rrp6. Signals were normalized to *act1+* mRNA levels and expressed relative to the wild type strain. Error bars represent the standard deviation from at least three independent experiments. Stars denote statistical significance relative to wild type cells (t-test p-values: *mei4+* = 2.4E-5; *ssm4+* = 7.84E-3; *mcp5+* = 5.88E-3). (**B**) RT-qPCR analysis of the *mei4+*, *ssm4 + and *mcp5+* meiotic transcripts in cells grown in minimal medium (EMM0.5X) and deleted for all non-essential Ccr4-Not subunits (i.e. all but Not1) or Rrp6 as a control. Signals were normalized to *act1+* mRNA levels and expressed relative to the wild type strain. Error bars represent the standard deviation from at least three independent experiments. Stars denote statistical significance relative to wild type cells (t-test p-values for *mot2Δ* and *rrp6Δ* strains: *mei4+* = 2.28E-3 and 4.34E-3; *ssm4+* = 1.74E-3 and 1.34E-3; *mcp5+* = 2.26E-3 and 5.83E-3). (**C**) Comparison of the *mot2Δ* (duplicate) and wild type (triplicate) transcriptomes in minimal medium (EMM0.5X). Volcano plot shows the fold change (log2) on the x axis and the P-value distribution (-log10 P value) on the y axis for the transcripts identified in RNA-seq analysis. Each dot represents one transcript and the colour code refers to the different functional categories of Mmi1 targets, as described in *Hiriart et al. (2012)*. (**D**) Venn diagram showing the overlap of transcripts stabilized in *mot2Δ* and *rrp6Δ* strains and compared to Mmi1 targets (*Hiriart et al., 2012*). The R package «SuperExactTest» (*Wang et al., 2016*) was used to calculate the p-values of intersects: *mot2Δ ∩ rrp6Δ* = 6.98E-24; *mot2Δ ∩ mmi1Δ* = 4.4E-25; *mmi1 Δ ∩ rrp6Δ* = 1.63E-16; *mot2Δ ∩ rrp6Δ ∩ mmi1Δ* = 4.85E-40.

DOI: https://doi.org/10.7554/eLife.28046.005

*Figure 2 continued*

The following source data and figure supplement are available for figure 2:

**Source data 1.** Analysis of RNA-sequencing data.

DOI: https://doi.org/10.7554/eLife.28046.007

**Figure supplement 1.** The E3 ubiquitin ligase Mot2 binds to meiotic mRNAs.

DOI: https://doi.org/10.7554/eLife.28046.006

meiotic mRNA suppression was still observed in the absence of Mot2 (*Figure 3D*). These data indicate that increased Mei2 protein levels cannot solely arise from altered transcription and/or stability of *mei2+* mRNAs in *mot2Δ* cells.

We also examined whether the control of Mei2 abundance involves additional Ccr4-Not components. Deletion of the non-essential subunits of the complex other than Mot2, including the RNA deadenylases Ccr4 and Caf1, did not result in the accumulation of Mei2 (*Figure 3—figure supplement 1B*). This supports the notion that the regulation of Mei2 levels depends specifically on the integrity of the E3 ubiquitin ligase.

## Mot2 functions with Mmi1 to limit the accumulation of Mei2 and meiotic mRNAs

Our results strongly suggest that Mot2 suppresses the expression of meiotic RNAs in vegetative cells by acting on the Mmi1 inhibitor Mei2. It is however possible that Mot2 affects the levels of meiotic RNAs independently of Mei2. To demonstrate that the two events are causally linked, we assessed the levels of meiotic transcripts in a *mot2Δ mei2Δ* double mutant. Crucially, deletion of *mei2+* restored degradation of meiotic mRNAs in the absence of Mot2 but not in *rrp6Δ* cells (*Figure 4A*), indicating that loss of Mei2 does not induce degradation of these transcripts by another pathway. Importantly, over-expression of the *mei2+* gene in a wild type background led to a strong stabilization of meiotic RNAs (*Figure 4B*), phenocopying the effect of a *mot2Δ* mutant.

Thus, these results demonstrate that Mot2 is important to maintain low levels of Mei2 and/or repress its activity to ensure efficient Mmi1- and exosome-dependent degradation of meiotic transcripts during vegetative growth.

Examination of exponentially growing cells by microscopy did not reveal the occurrence of ectopic meiosis in *mot2Δ* cells (*Figure 4—figure supplement 1A*), supporting the notion that meiotic transcripts accumulate in actively dividing cells and not in a fraction of cells undergoing meiosis. *mot2Δ* cells displayed a moderate growth defect in minimal medium that was in part due to the inappropriate expression of meiotic RNAs since the defect was partially suppressed by deletion of *mei2+* (*Figure 4—figure supplement 1B*). Consistent with this notion, although the *mot2Δ* mutant also grew slower than wild type cells in rich medium, this defect was Mei2-independent (*Figure 4—figure supplement 1B*) and presumably due to other functions of Mot2.

According to the model that Mmi1 recruits Mot2 to limit the levels of Mei2 or control its activity, the association of Mei2 with Mmi1 should persist in the absence of the E3 ligase. Consistent with this notion, co-immunoprecipitation assays and MS analyses of Mmi1 purified from the *mot2Δ* mutant grown in minimal medium indicated that Mei2 persisted in association with Mmi1, as anticipated from its higher steady state levels (*Figure 4C*, *Figure 1—source data 1*).

Another prediction is that the control of Mei2 levels should depend on Mmi1. We indeed observed an accumulation of Mei2 in an *mmi1Δ* mutant, despite the presence of wild type Mot2 (*Figure 4D*). Instead, neither Red1 nor Rrp6 affected Mei2 levels. This indicates that Mmi1 and Mot2 are components of a cellular pathway that is distinct from the post-transcriptional regulation of meiotic mRNAs exerted by Mmi1, MTREC and the exosome.

Recent work showed that the most N-terminal residues of Mmi1 are required for its association with the Ccr4-Not complex (*Stowell et al., 2016*). Expression of a variant of Mmi1 lacking the first 65 amino acids (Mmi1-(*1-65*)Δ) led to the accumulation of Mei2, similarly to *mot2Δ* cells (*Figure 4E*), strongly suggesting that the recruitment of Ccr4-Not by Mmi1 is required to maintain low levels of Mei2.

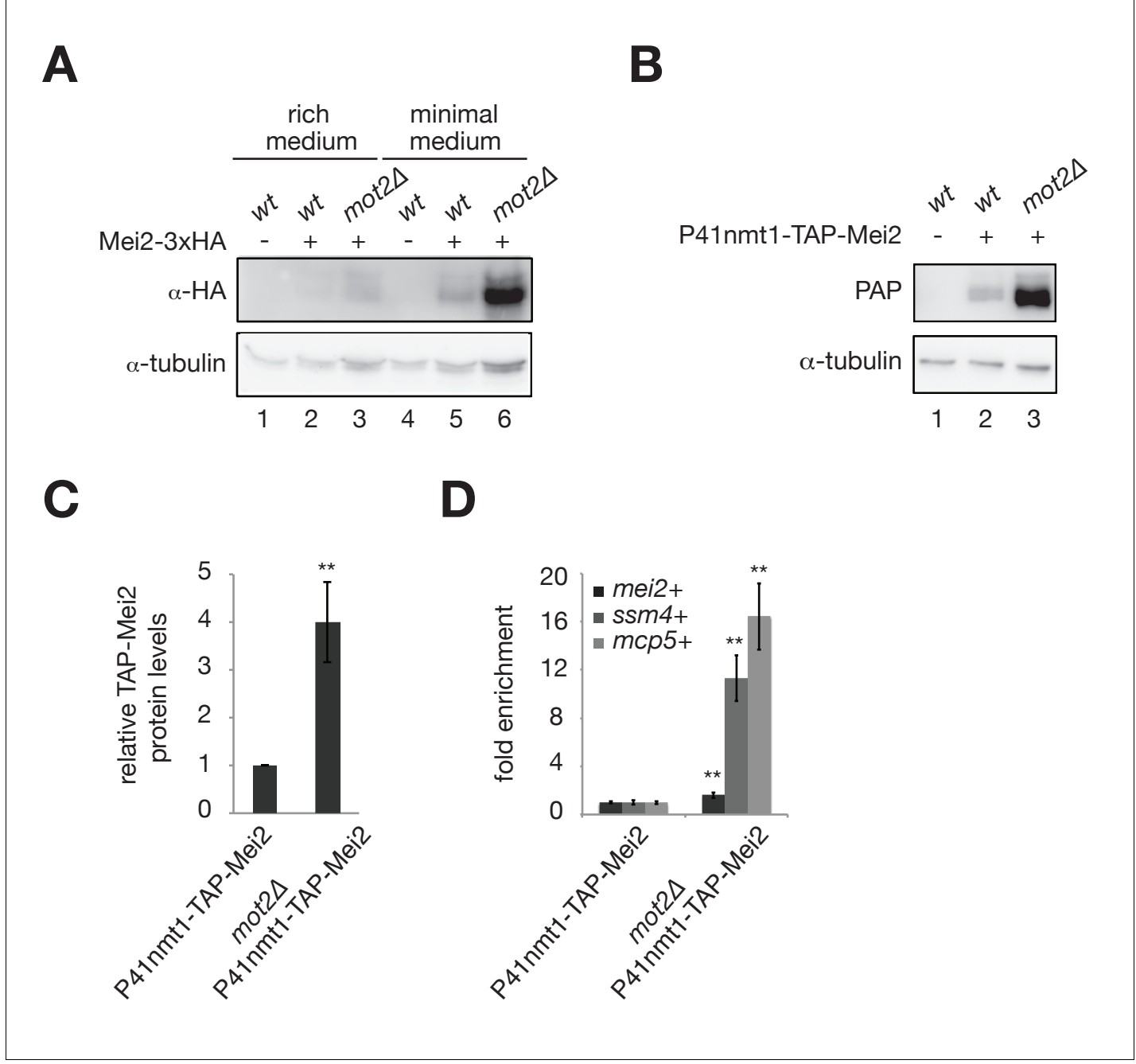

**Figure 3.** The E3 ubiquitin ligase Mot2 negatively affects the levels of the Mmi1 inhibitor Mei2. (A) Western blot showing total Mei2-3xHA levels in wt and *mot2Δ* cells grown at 30°C to mid-log phase in rich (YE) and minimal (EMM0.5X) media. An anti-tubulin antibody was used as a loading control. (B) Western blot showing total TAP-Mei2 levels expressed from the P41nmt1 promoter in wt and *mot2Δ* cells grown in minimal medium (EMM0.5X). An anti-tubulin antibody was used as a loading control. (C) Quantification of TAP-Mei2 protein levels, normalized to tubulin and expressed relative to wt cells. Error bars represent the standard deviation from five independent experiments. Stars denote statistical significance (t-test p-value=1.38E-3). (D) RT-qPCR analysis of *mei2+*, *ssm4+* and *mcp5+* transcripts in wt and *mot2Δ* strains expressing Mei2 from the P41nmt1 promoter. Cells were grown in minimal medium (EMM0.5X). Signals were normalized to *act1+* mRNA levels and expressed relative to the P41nmt1-TAP-Mei2 strain. Error bars represent the standard deviation from four independent experiments. Stars denote statistical significance (t-test p-values: *mei2+* = 8.5E-3; *ssm4+* = 1.53E-3; *mcp5+* = 1.49E-3).

DOI: https://doi.org/10.7554/eLife.28046.008

The following figure supplement is available for figure 3:

**Figure supplement 1.** The E3 ubiquitin ligase Mot2 negatively affects the levels of the Mmi1 inhibitor Mei2.
DOI: https://doi.org/10.7554/eLife.28046.009

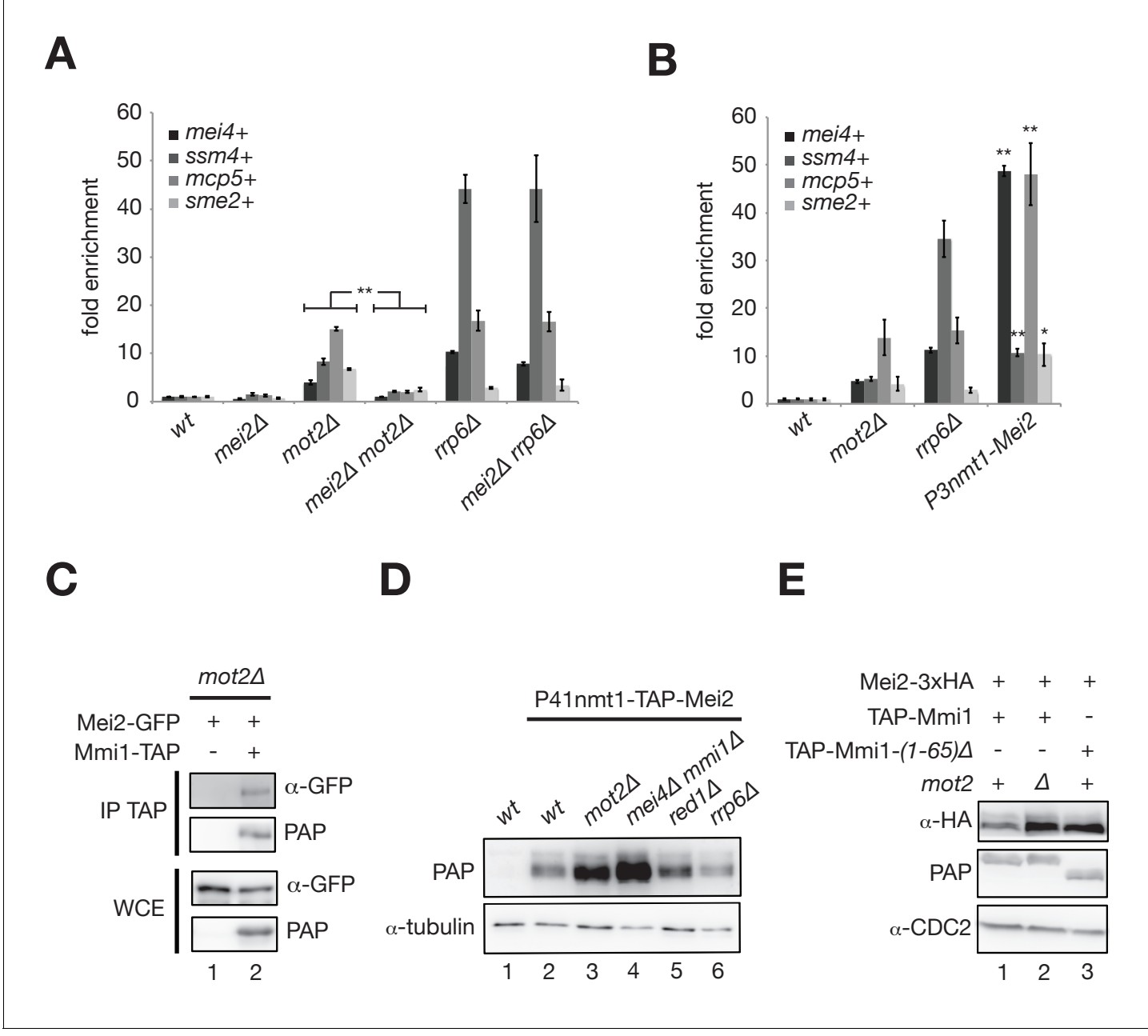

**Figure 4.** Mot2 functions with Mmi1 to limit the accumulation of Mei2 and meiotic mRNAs. (A), (B) RT-qPCR analyses of meiotic transcripts in cells of the indicated genetic backgrounds and grown in minimal medium (EMM0.5X). Shown is the fold enrichment of RNAs levels normalized to *act1* + transcripts and expressed relative to the wild type strain. Error bars represent the standard deviation from three independent experiments. (A) Pairwise comparisons of four meiotic transcripts in *mot2Δ* versus *mei2Δ mot2Δ* mutants give p-values<7.3E-3. (B) Stars denote statistical significance relative to wild type cells (t-test p-values: *mei4+* = 1.29E-4; *ssm4+* = 1.9E-3; *mcp5+* = 6.3E-3; *sme2+* = 2.04E-2). (C) Western blot showing that Mei2-GFP co-immunoprecipitates with Mmi1-TAP upon RNaseA/T1 treatment in *mot2Δ* cells grown in minimal medium (EMM0.5X). (D), (E) Western blots showing total Mei2 levels (P41nmt1-TAP-Mei2 in (D) or Mei2-3xHA in (E)) in cells of the indicated genetic backgrounds and grown in minimal medium (EMM0.5X). Anti-tubulin and anti-CDC2 antibodies were used as loading controls in panels (D) and (E), respectively.

DOI: https://doi.org/10.7554/eLife.28046.010

The following figure supplement is available for figure 4:

**Figure supplement 1.** Phenotypic characterization of the *mot2Δ* mutant.
DOI: https://doi.org/10.7554/eLife.28046.011

## Mot2 and Mmi1 do not repress *mei2+* mRNA translation

The Ccr4-Not complex has been involved in mRNA translational repression (*Miller and Reese, 2012*; *Collart, 2016*). To determine whether Mot2 affects the production of Mei2, we analyzed the levels of *mei2+* mRNAs that co-immunoprecipitate with the translation machinery in wild type and *mot2Δ* cells. If Mot2 represses translation of *mei2+* mRNAs, then the fraction of transcripts associated with translating ribosomes should increase in the *mot2Δ* mutant. Contrary to this scenario, both the 60S ribosomal subunit Rpl1601 and the elongation factor Tef3 did not pull down more *mei2+* transcripts in the absence of Mot2, despite identical protein levels (*Figure 5A*, and *Figure 5—figure supplement 1A,C*). Further indicating that *mei2+* mRNA translation is not specifically increased in *mot2Δ* cells, the fraction of immunoprecipitated *mei2+* mRNAs relative to *act1+* transcripts was even slightly decreased in the mutant (*Figure 5B*, and *Figure 5—figure supplement 1B*).

Since Mmi1 interacts with the Ccr4-Not complex (*Figure 1*; *Cotobal et al., 2015*; *Ukleja et al., 2016*; *Sugiyama et al., 2016*; *Stowell et al., 2016*), we also investigated whether it associates with *mei2+* transcripts. We found that Mmi1 co-immunoprecipitates DSR-containing transcripts (e.g. *mei4+* and *mcp5+*) but not *mei2+* mRNAs, further supporting the notion that Mmi1 and Mot2 do not partake in the translational control of Mei2 (*Figure 5C*).

Finally, we compared Mei2 protein levels in strains carrying the *mts2-1* thermosensitive mutation of the proteasome subunit Rpt2/Mts2 alone or in combination with the deletion of *mot2+*. If the absence of Mot2 leaded to enhanced translation of *mei2+* transcripts, then Mei2 levels should be further increased in an *mts2-1 mot2Δ* double mutant relative to a *mts2-1* single mutant because of increased production and decreased degradation. Conversely, if Mot2 and the proteasome function in the same degradation pathway, Mei2 levels should not be affected by deletion of *mot2+* in *mts2-1* cells. Consistent with the latter notion, Mei2 accumulates to similar levels in both *mts2-1* and *mts2-1 mot2Δ* mutants at the non-permissive temperature, suggesting that Mot2 triggers Mei2 degradation via the proteasome pathway (*Figure 5D*, compare lanes 3 and 5). Note that lower-migrating Mei2 isoforms were specifically detected in *mts2-1* cells, possibly reflecting the accumulation of phosphorylated and/or ubiquitinated degradation intermediates.

Together, these results support the notion that Mmi1 and Mot2 do not mediate translational repression of *mei2+* mRNAs.

## The E3 ubiquitin ligase activity of Mot2 is required for repressing Mei2 during vegetative growth

Because Mot2 is an E3 ubiquitin ligase, we assessed whether its catalytic activity is required for controlling Mei2 during vegetative growth. Expression of Mot2 variants lacking the RING domain (Mot2-RINGΔ) or carrying single substitutions in residues involved in substrate ubiquitination (Mot2-C37A, -C45A and -C57A) did not restore wild type levels of Mei2 in a *mot2Δ* background (*Figure 6A*). This indicates that the ubiquitination activity of Mot2 is required for suppressing expression of Mei2 in vegetative cells.

To address whether Mei2 is a substrate for ubiquitination by Mot2 in vivo, we purified total cellular ubiquitinated proteins from cells expressing $His_6$-tagged ubiquitin. The presence of ubiquitinated species of C-terminally 3xHA-tagged Mei2 was probed by immunoblotting in wild type and *mot2Δ* cells. Because total Mei2 levels are higher in the absence of Mot2, we compared the levels of Mei2 ubiquitinated species (Ubi-Mei2) relative to the total in the two strains. Up to four specific bands corresponding to Ubi-Mei2 could be observed in wild type cells (*Figure 6B*, lane 6). Deletion of *mot2+* resulted in the detection of similar levels of Ubi-Mei2 in spite of the higher amount of total Mei2, which corresponds to an approximate four-fold decrease in the fraction of ubiquitinated species (*Figure 6C*; see also *Figure 6—figure supplement 1A* for equal Mei2 loading). We also analyzed the fraction of Ubi-Mei2 species in *mts2-1* and *mts2-1 mot2Δ* mutants, in which the levels of total Mei2 are nearly identical, and reproducibly observed a 2-fold decrease in the absence of Mot2 in this context (*Figure 6—figure supplement 1B–C*).

To exclude the possibility that the decrease in the fraction of Ubi-Mei2 species in the *mot2Δ* mutant is the consequence - and not the cause - of increased Mei2 levels, we compared the fraction of Ubi-Mei2 in cells expressing endogenous levels of the protein (e.g. Mei2-3xFLAG) or overexpressing it (e.g. P3nmt1-3xFLAG-Mei2). Even upon a roughly 100-fold Mei2 overexpression, we observed a proportional increase in Ubi-Mei2 levels (*Figure 6—figure supplement 1D–E*). This indicates that

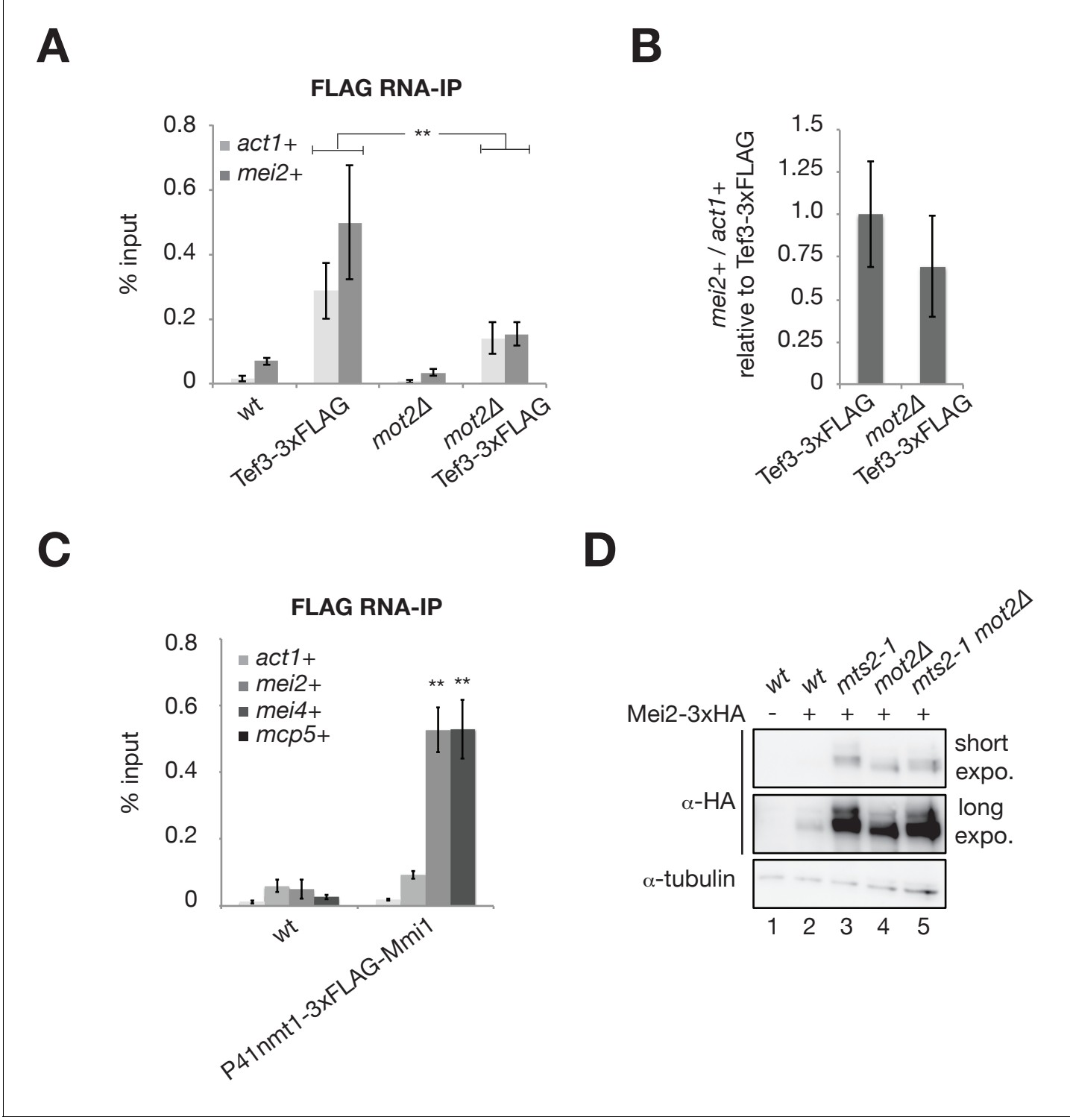

**Figure 5.** Mot2 and Mmi1 do not repress mei2+ mRNA translation. (**A**) RNA-immunoprecipitation experiments in wild type and *mot2Δ* cells. Shown are the enrichments (% input) of *act1+* and *mei2+* mRNAs upon pulldown of the 3xFLAG-tagged translation elongation factor Tef3. Error bars represent the standard deviation of six independent immunoprecipitations from at least three biological replicates. Stars denote statistical significance between samples (t-test p-values: *act1+* = 6.87E-3; *mei2+* = 4.26E-3). (**B**) Quantification of *mei2+* mRNA levels normalized to *act1+* transcripts and expressed relative to the wild type tagged strain (Tef3-3xFLAG). Error bars represent the standard deviation of six independent immunoprecipitations from at least three biological replicates. (**C**) RNA-immunoprecipitation experiments in wild type cells. Shown are the enrichments (% input) of *act1+*, *mei2+*, *mei4+* and *mcp5+* transcripts upon pulldown of 3xFLAG-tagged Mmi1. Error bars represent the standard deviation of four independent

*Figure 5 continued on next page*

**eLIFE** Research article

Genes and Chromosomes

*Figure 5 continued*

immunoprecipitations from three biological replicates. Stars denote statistical significance (t-test p-values: *mei4+* = 1.87E-4; *mcp5+* = 1.35E-3). (D) Western blot showing total Mei2-3xHA levels in cells of the indicated genetic backgrounds grown in minimal medium (EMM0.5X) following a temperature shift at 37°C for 1 hr. Shown are short and long exposures for the anti-HA immunoblotting. An anti-tubulin antibody was used as a loading control.

DOI: https://doi.org/10.7554/eLife.28046.012

The following figure supplement is available for figure 5:

**Figure supplement 1.** Mot2 does not repress *mei2+* mRNA translation.

DOI: https://doi.org/10.7554/eLife.28046.013

ubiquitination is not limiting and that the decreased level of Ubi-Mei2 in *mot2Δ* cells is not the mere consequence of an overproduction of Mei2.

Together, these results are consistent with the notion that Mot2 ubiquitinates Mei2, which likely promotes its degradation and/or inhibits its activity.

Previous work showed that mutation of the E3 ubiquitin ligase Ubr1 also triggers accumulation of Mei2 in vegetative cells (*Kitamura et al., 2001*). To determine which of the two E3 ligases is responsible for the main Mei2 turnover pathway, we compared total and Ubi-Mei2 levels in *ubr1Δ* and *mot2Δ* cells. Total Mei2 levels were strongly increased in the absence of Ubr1 relative to the *mot2Δ* mutant (*Figure 6B*, *Figure 6—figure supplement 2A–B*). This was due to a major effect on the turnover rate of Mei2, as shown by cycloheximide chase experiments (*Figure 6—figure supplement 2C*). Note that because of the low levels of Mei2 in wild type cells and the relatively modest impact of Mot2 on Mei2 stability, it is not possible to reliably evaluate the differences in turnover rate between the wild type and *mot2Δ* strains (see discussion). Consistent with a predominant role for Ubr1, ubiquitination of Mei2 was strongly decreased in *ubr1Δ* cells (*Figure 6B–C*). Live-cell microscopy of GFP-tagged Mei2 further substantiated the major requirement for Ubr1 in the control of Mei2 abundance (*Figure 6—figure supplement 2D*). Finally, increased Mei2 levels in *ubr1Δ* cells resulted in a pronounced accumulation of Mmi1-targeted meiotic mRNAs (*Figure 6D*), highlighting the functional importance of suppressing Mei2 levels.

From these results, we conclude that Mot2 is not responsible for the main turnover pathway of Mei2, which depends on Ubr1. We propose that the control exerted by Mot2 on Mei2 has an important role in fine-tuning its levels and/or regulating its function, allowing full repression of meiotic mRNAs during vegetative growth (*Figure 7*).

## Discussion

Fission yeast cells selectively eliminate meiosis-specific transcripts during the mitotic cell cycle to inhibit sexual differentiation and ensure robust vegetative growth. Here, we report that the E3 ubiquitin ligase subunit Not4/Mot2 of the Ccr4-Not complex strengthens the control exerted by the YTH-family RNA-binding protein Mmi1 on the repression of the meiotic program during vegetative growth (*Figure 7*). Our results suggest that Mmi1 recruits Mot2 to ubiquitinate its own inhibitor Mei2, although we cannot formally exclude that the ubiquitination activity of Mot2 impacts the function or levels of Mei2 in an indirect manner. Importantly, Mot2 is not involved in the constitutive pathway of Mei2 turnover, which depend on the E3 ubiquitin ligase Ubr1, but rather targets a fraction of Mei2 to prevent inhibition of Mmi1. This ensures the maintenance of Mmi1 in a functional state and the persistent suppression of meiotic mRNAs. Thus, Mmi1 has a dual role: in nuclear RNA surveillance, by targeting meiotic transcripts for degradation by the exosome, and in the regulation of protein ubiquitination, by recruiting Mot2 to Mei2. These results also reveal a novel role for the E3 ubiquitin ligase Mot2 in the control of sexual differentiation in fission yeast.

The existence of a regulatory circuit whereby a protein controls the levels or the function of its own inhibitor might hamper alterations in the activity of the protein (Mmi1) due to fluctuations in the levels of the inhibitor (Mei2), which might occur under given growth conditions. This, in turn, is expected to prevent variations in the levels of meiotic RNAs and their translation products, which might affect, directly or indirectly, the robustness of vegetative growth.

The need for buffering increased Mei2 production is the likely reason why Mot2 only affects the levels of meiotic transcripts during growth in minimal medium. In yeast, meiosis occurs upon

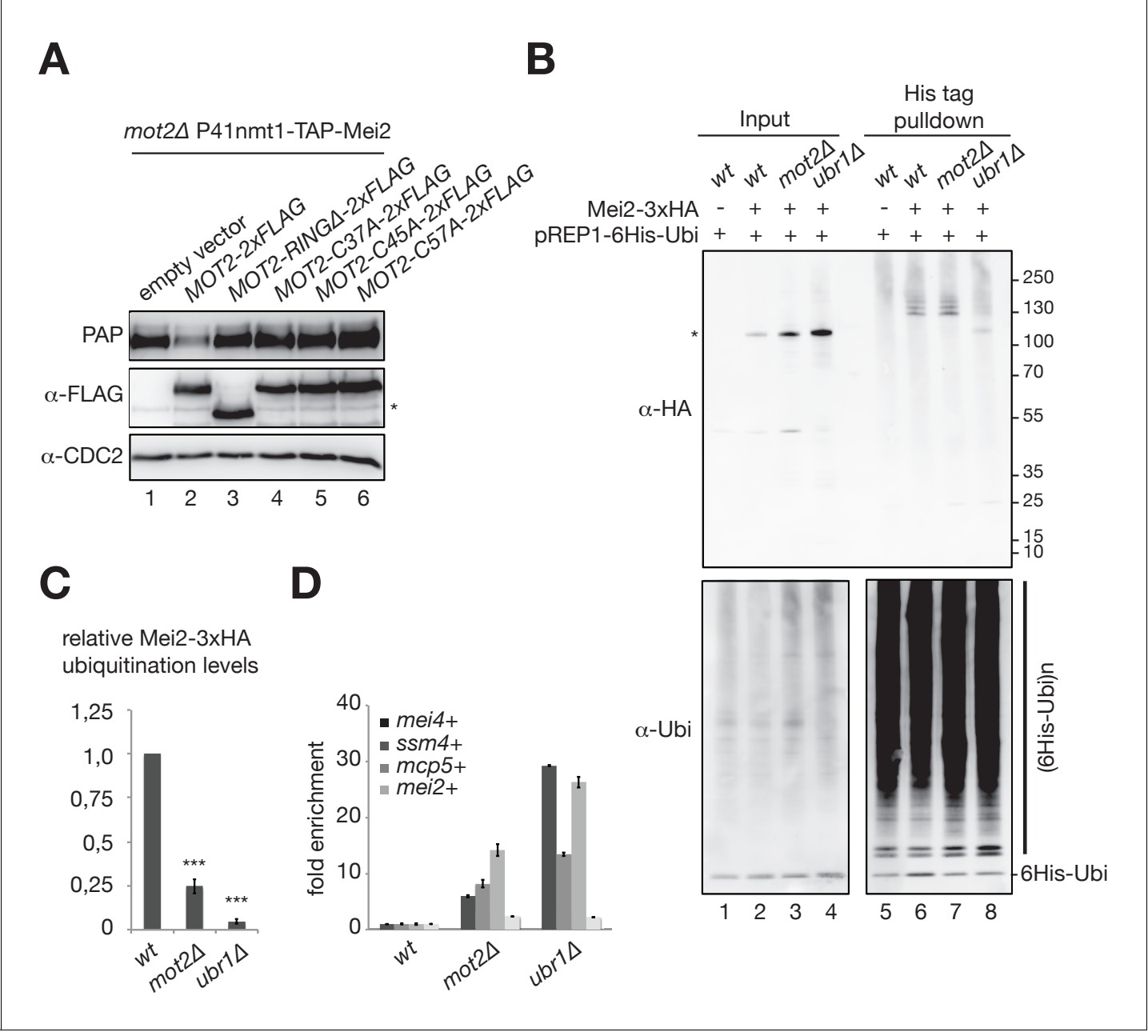

**Figure 6.** The E3 ubiquitin ligase activity of Mot2 is required for repressing Mei2 during vegetative growth. (**A**) Western blot showing total TAP-Mei2 levels expressed from the P41nmt1 promoter in cells of the indicated genetic backgrounds and grown in minimal medium lacking leucine (EMM-LEU0.5X). An anti-FLAG antibody was used to detect Mot2 variants expressed from the pREP41 vector. An anti-CDC2 antibody was used as loading control. The star denotes a non-specific band. (**B**) In vivo ubiquitination of Mei2-3xHA in wt, *mot2Δ* and *ubr1Δ* cells expressing 6His tagged-ubiquitin in minimal medium (EMM-LEU0.5X). Total and ubiquitinated Mei2 as well as ubiquitin conjugates were analyzed by immunoblotting using anti-HA and anti-ubiquitin antibodies respectively. An untagged wild type strain was used as negative control. The star denotes unmodified Mei2 molecules. (**C**) Quantification of Mei2 ubiquitinated species relative to total protein levels and expressed relative to Mei2-3xHA wild type cells. Error bars represent the standard deviation from five independent experiments. Stars denote statistical significance relative to Mei2-3xHA wild type cells (t-test p-values=2.29E-6 for *mot2Δ*, and 5.64E-9 for *ubr1Δ*). (**D**) RT-qPCR analysis of meiotic transcripts in wt, *mot2Δ* and *ubr1Δ* cells. Shown is the fold enrichment of RNAs levels normalized to *act1+* transcripts and expressed relative to the wild type strain. Error bars represent the deviation from the mean of biological duplicates.

DOI: https://doi.org/10.7554/eLife.28046.014

The following figure supplements are available for figure 6:

**Figure supplement 1.** Contribution of Mot2 to the ubiquitination of Mei2.

*Figure 6 continued on next page*

*Figure 6 continued*

DOI: https://doi.org/10.7554/eLife.28046.015

**Figure supplement 2.** Both Mot2 and Ubr1 limit the accumulation of Mei2.

DOI: https://doi.org/10.7554/eLife.28046.016

exposure to nutritional starvation, which activates signalling pathways that converge towards the induction of a complex transcriptional program finalized to sexual differentiation (*Yamamoto, 2010*). Growth in minimal medium, when nutrients are available but limiting, might induce a partially analogous response, possibly anticipating the need for entry into meiosis. Indeed, we observed increased levels of Mei2 protein upon vegetative growth in minimal versus rich medium (*Figure 3A*), which might in principle be due to increased transcription, translation and/or protein stabilization. The low levels of Mei2 in rich medium even upon deletion of *mot2*+ suggest that the expression of the *mei2* + gene is limited by transcription in these growth conditions. Conversely, the requirement for Mot2 ubiquitination activity to restrict Mei2 levels or activity in minimal medium (*Figure 6A*) might suggest that Mei2 is post-translationally controlled in this context. Thus, Mot2 has an important role in controlling the levels of Mei2 under conditions where entry into meiosis is to some extent transcriptionally prefigured, but for which a no-return commitment is premature.

Higher levels of Mei2 in *mot2Δ* cells could be due to increased Mei2 production, consistent with a known role of the Ccr4-Not complex in translational repression (*Miller and Reese, 2012*; *Collart, 2016*). However, our data argue against this possibility because (i) the fraction of *mei2*+ mRNAs associated with translating ribosomes does not increase in the absence of Mot2 (*Figure 5A–B*, *Figure 5—figure supplement 1A–B*), (ii) the regulation of Mei2 by Mot2 requires Mmi1 (*Figure 4D–E*), which localizes to the nucleus (*Harigaya et al., 2006*; *Sugiyama and Sugioka-Sugiyama, 2011*;

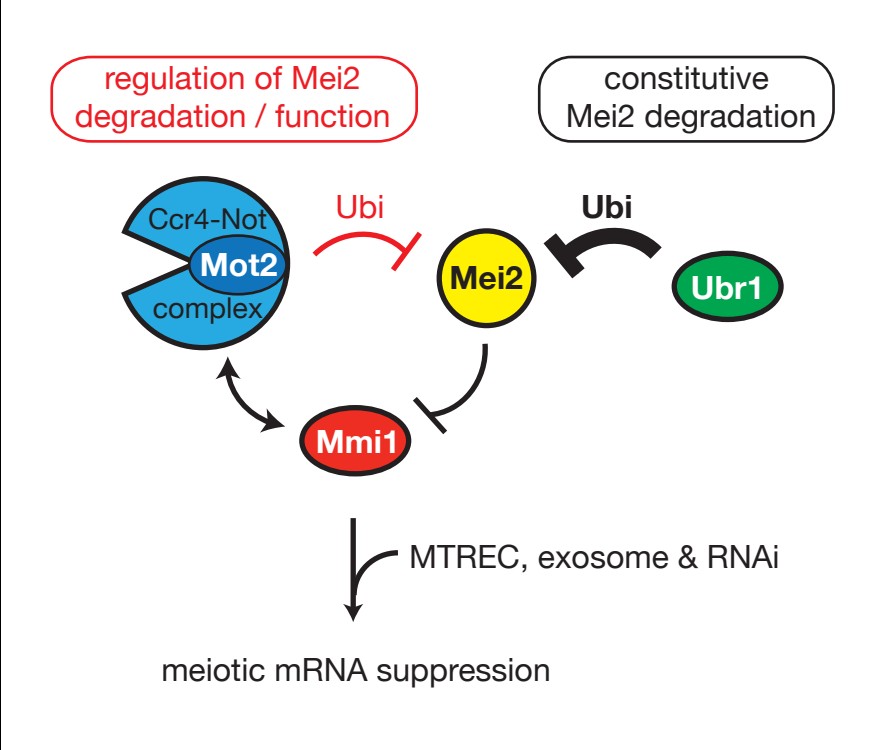

**Figure 7.** Model. Shown is the regulatory circuit whereby Mmi1 recruits Mot2, likely in the context of Ccr4-Not, to ubiquitinate its own inhibitor Mei2. The E3 ubiquitin ligase Ubr1 is involved in the constitutive pathway of Mei2 turnover. Both Mot2 and Ubr1 contribute to maintain low levels of Mei2, thereby preserving the activity of Mmi1 and sustaining meiotic mRNAs suppression in vegetative cells.

DOI: https://doi.org/10.7554/eLife.28046.017

*Yamashita et al., 2013*; *Shichino et al., 2014*; *Sugiyama et al., 2016*) and does not bind to *mei2* + mRNAs (*Figure 5C*), and (iii) the absence of Mot2 does not lead to further increased Mei2 levels upon proteasome inactivation (*Figure 5D*). Our results also exclude significant effects of Mot2 on *mei2*+ transcription and mRNA stability (*Figure 3B–D*). Together with the demonstration that Mot2 ubiquitination activity is required for controlling Mei2 protein levels (*Figure 6A*), these results suggest that mutation of Mot2 affects the stability of Mei2.

The turnover of Mei2 mainly depends on the E3 ubiquitin ligase Ubr1. Considering the steady state levels of Mei2 in *ubr1Δ* and *mot2Δ* cells, the Mot2-dependent degradation rate is expected to be approximately six times less active than that dependent on Ubr1 (*Figure 6—figure supplement 2A–B*). This implies that Mei2 half-life might differ only by roughly 15% between wild type (in which both ubiquitin ligases are active) and *mot2Δ* cells (in which only Ubr1 is active), which amounts to less than 1.5 min. This small difference, together with the very low levels of Mei2 in wild type cells, is the main reason why we could not reliably measure the impact of Mot2 deletion on Mei2 degradation rate. Because of this limitation, we cannot formally exclude the possibility that indirect effects may account for the increased Mei2 steady state levels in the absence of Mot2. Yet, we showed that higher Mei2 levels in *mot2Δ* cells did not result from increased transcription and translation, pointing to the most parsimonious explanation that Mot2 impacts directly Mei2 turnover.

It is however unclear whether maintaining the levels of Mei2 under a given threshold is the main physiological role of Mot2 for controlling its activity or whether ubiquitination by itself is the key regulatory event. In the latter scenario, ubiquitination would be the primary Mei2 inhibitory event and Mot2-dependent protein degradation would ensue as a secondary consequence.

In vegetative cells, Mmi1 localizes to scattered nuclear foci to mediate meiotic RNAs degradation (*Sugiyama and Sugioka-Sugiyama, 2011*; *Yamashita et al., 2013*; *Shichino et al., 2014*; *Sugiyama et al., 2016*). It is possible that Mot2 ubiquinates Mei2 to prevent its entry and/or its persistence in nuclear dots, thereby preserving the function of Mmi1. Ubiquinated Mei2 might then be directed more efficiently to the turnover pathway, thus explaining the impact of Mot2 on Mei2 steady state levels.

The control exerted by Mot2 (and Ubr1) on Mei2 levels is only part of the mechanism that prevents meiosis onset in vegetative cells. Although Mei2 levels in *mot2Δ* cells exceed those observed in the wild type strain in minimal medium (*Figure 3A*), this is not sufficient to induce ectopic entry in meiosis (*Figure 4—figure supplement 1A*). This observation suggests that additional events are required to activate meiosis and/or that another control exists on Mei2 that backs up the roles of the E3 ligases. Consistent with this latter hypothesis, the Pat1 and Tor2 protein kinases phosphorylate Mei2 to inhibit its activity and promote its degradation, and mutation of these enzymes or expression of a non-phosphorylatable form of Mei2 trigger ectopic entry into meiosis, even in haploid cells (*Yamamoto, 2010*; *Kitamura et al., 2001*; *Alvarez and Moreno, 2006*; *Otsubo et al., 2014*). Thus, it is possible that the increased pools of Mei2 in *mot2Δ* and *ubr1Δ* cells might be partially inactivated by Pat1 and/or Tor2, thereby preventing full activation of the meiotic program.

A strict control on the inhibition of Mmi1 appears to be essential given the important role of this factor in suppressing mRNAs encoding meiosis-specific transcription factors that, in turn, control the expression of other transcription factors (*Mata et al., 2007*). On the other side, the onset of meiosis might require abrupt changes in physiology, which could benefit from a faster response associated to releasing the inhibition of factors already present in the cell as opposed to synthesizing these factors de novo. Whether this also requires regulation of expression or activity of Mot2 is an important question and matter for future studies.

The function of Ccr4-Not in sexual differentiation in fission yeast is analogous to the role played by the metazoan complex in controlling developmental decisions (*Rouget et al., 2010*; *Joly et al., 2013*; *Nousch et al., 2013*; *Solana et al., 2013*; *Yamaji et al., 2017*). However, in these examples, the mechanism of action of the complex essentially relies on its deadenylation and translational repression activities. The YTH proteins Pho92 and YTHDF2, the budding yeast and human homologues of Mmi1 respectively, also recruit Ccr4-Not to regulate mRNA deadenylation and turn over (*Kang et al., 2014*; *Du et al., 2016*). In light of these findings, the control of Mei2 abundance by Mmi1 and the conserved E3 ubiquitin ligase Not4/Mot2 might provide another layer of complexity to the regulatory potential of the Ccr4-Not complex. Our work paves the way for addressing the conservation of this type of regulation in other networks and species.

## Materials and methods

### Fission yeast strains and growth media

The *S. pombe* strains used in this study are listed in *Supplementary file 1*. Strains were generated by transformation with a lithium acetate-based method or by random spore analysis. *mmi1Δ* cells were generated from a parental strain possessing a deletion of *mei4+*, since the absence of Mmi1 leads to severe growth and viability defects due to the deleterious expression of Mei4, a key meiosis-specific transcription factor. Growth media included complete medium (YE Broth, Formedium, #PMC0105) and minimal medium (EMM Broth, Formedium, #PMD0210). All experiments in minimal medium were performed using EMM 0.5X supplemented with 1% glucose (2% final concentration) and 125 mg/L of each adenine (Sigma, #A2786), L-histidine (Sigma, #H8000), uracil (Sigma, #U750), L-lysine (Sigma, #L5501) and L-leucine (Sigma, #L8000).

### Affinity purification and mass spectrometry analysis

500 mL of cells were grown at 30°C in EMM 0.5X until $OD_{600nm}$ = 1.0–1.2 and harvested by centrifugation. Cell pellets were resuspended in 5 mL lysis buffer (6 mM $Na_2HPO_4$, 4 mM $NaH_2PO_4$, 150 mM $NaC_2H_3O_2$, 5 mM $MgC_2H_3O_2$, 0.25% NP-40, 2 mM EDTA, 1 mM EGTA, 5% glycerol, 1 mM AEBSF, 4 mM benzamidine and 2X Roche complete EDTA-free protease inhibitor cocktail) and slowly dropped into liquid nitrogen to form 'pop-corn'. Lysis was performed using a Ball Mill (Retsch, MM400) for 15 min at a 10 Hz frequency. Extracts were cleared by centrifugation before precipitation with 7.5 mg of pre-washed rabbit IgG-conjugated M-270 Epoxy Dynabeads (Invitrogen, #14311D) for 1 hr at 4°C. Lysates were incubated with or without 1 µL RNaseA/T1 cocktail (Ambion, #AM2286) per mL of extract for 20 min at 4°C prior to immunoprecipitation. Beads were then washed twice with IPP150 (10 mM Tris pH8, 150 mM NaCl, 0.1% NP-40). Immunoprecipitated complexes were eluted over-night at 4°C by adding 50 units TEV protease (Invitrogen, #12575–015) in 200 µL TEV cleavage buffer (10 mM Tris pH8, 150 mM NaCl, 0.1% NP-40, 0.5 mM EDTA, 1 mM DTT). 5% of eluates were subjected to silver staining using the SilverQuest Silver Staining kit (Invitrogen, #LC6070) and the remaining samples were precipitated with methanol-chloroform. Briefly, TEV eluates were mixed sequentially with 4 volumes of methanol, 1 vol of chloroform and 3 volumes of water. Samples were centrifuged at 16000 g for 30 min at 4°C and the upper phases were discarded. 3 volumes of methanol were added to the lower phases to extract chloroform and samples were vortexed, left at −20°C for 30 min and centrifuged at 16000 g for 20 min at 4°C. Following removal of supernatants, protein precipitates were air dried and stored at −80°C prior to MS/MS analysis.

Precipitated proteins were digested overnight at 37°C by sequencing grade trypsin (12.5 µg/ml; Promega Madison, Wi, USA) in 20 µl of $NH_4HCO_3$ 25 mmol/L. Digests were analyzed by an Orbitrap Fusion Tribrid mass spectrometer (Thermo Fisher Scientific, San Jose, CA) equipped with a Thermo Scientific EASY-Spray nanoelectrospray ion source and coupled to an Easy nano-LC Proxeon 1000 system (Thermo Fisher Scientific, San Jose, CA). Chromatographic separation of peptides was performed with the following parameters: pre-column Acclaim PepMap100 (2 cm, 75 µm i.d., 3 µm, 100 Å), column Pepmap-RSLC Proxeon C18 (50 cm, 75 µm i.d., 2 µm, 100 Å), 300 nl/min flow, gradient rising from 95% solvent A (water, 0.1% formic acid) to 35% B (100% acetonitrile, 0.1% formic acid) in 98 min. Peptides were analyzed in the orbitrap in full ion scan mode at a resolution of 60000 (at *m/z* 400) and with a mass range of *m/z* 350–1550. Fragments were obtained with a Higher-energy Collisional Dissociation (HCD) activation with a collisional energy of 30%, and a quadrupole isolation width of 1.6 Da. MS/MS data were acquired in the linear ion trap in top-speed mode, with a dynamic exclusion of 50 s and a repeat duration of 60 s. The maximum ion accumulation times were set to 250 ms for MS acquisition and 60 ms for MS/MS acquisition in parallelization mode. MS/MS data were processed with Proteome Discoverer 1.4 software (Thermo Fisher scientific, San Jose, CA) coupled to an in house Mascot search server (Matrix Science, Boston, MA; version 2.5.1). The mass tolerance was set to 7 ppm for precursor ions and 0.5 Dalton for fragments. The following modifications were used in variable modifications: oxidation (M), phosphorylations (STY), acetylations (K, N-term), deamidations (N, Q), methylations (K), ubiquitinylation (GG and LRGG motifs on K amino acids). The maximum number of missed cleavages by trypsin was limited to 2. MS/MS data were searched against SwissProt databases with the *Schizosaccharomyces pombe* taxonomy. False

Discovery Rate (FDR) for peptides was calculated using the Percolator algorithm and peptides were considered identified under the 1% FDR threshold.

## Coimmunoprecipitation

CoIPs were performed essentially as described in the previous section with the following modifications: 50 ODs of cells were grown at 30°C in EMM 0.5X and harvested by centrifugation. Cell pellets were resuspended in 2 ml lysis buffer to make 'pop-corn'. 1 mg of pre-washed rabbit IgG-conjugated M-270 Epoxy Dynabeads (Invitrogen, #14311D) was used for immunoprecipitation in the presence or the absence of RNaseA/T1 cocktail (Ambion, #AM2286). Following washes, precipitates and input fractions were boiled at 95°C for 10 min in the presence of sample loading buffer and analyzed by SDS-PAGE and Western blotting using 1:3000 peroxydase-conjugated antiperoxydase (PAP) (Sigma, #P1291, RRID:AB_1079562), 1:3000 monoclonal anti-FLAG antibody (Sigma, #F3165, RRID: AB_259529), 1:3000 monoclonal anti-GFP antibody (Roche, #11814460001, RRID:AB_390913) and 1:4000 goat anti-mouse IgG-HRP (Santa Cruz Biotechnology, #sc-2005, RRID:AB_631736). Detection was performed using SuperSignal West Pico Chemiluminescent Substrate (ThermoFischer Scientific, #34080), ECL Select reagent (GE Healthcare, #RPN2235) and a Fujifilm LAS-4000 imager.

## Total protein analysis

2 to 5 ODs of cells grown at 30°C in YE, EMM 0.5X were harvested and lysed on ice in the presence of 0.3M NaOH and 1% β-mercaptoethanol for 15 min with occasional vortexing. Extracts were treated with TCA (7% final concentration) for 15 min on ice before full speed centrifugation at 4°C. Pellets were then resuspended in loading buffer (200 mM phosphate buffer pH 6.8, 8 M urea, 5% SDS, 1 mM EDTA, 100 mM DTT, 0.08% bromophenol blue) and heat-denatured at 70°C for 10 min. Soluble fractions were recovered and samples were analyzed by standard immunoblotting procedures using 1:3000 peroxydase-conjugated antiperoxydase (PAP) (Sigma, #P1291, RRID:AB_1079562), 1:3000 monoclonal anti-HA antibody (12CA5) (Sigma, #11583816001, RRID:AB_514505), 1:3000 monoclonal anti-FLAG antibody (Sigma, #F3165, RRID:AB_259529), 1:3000 monoclonal anti-tubulin antibody (Abcam, #ab6160, RRID:AB_305328), 1:3000 anti-CDC2 antibody (Abcam, #ab5467, RRID:AB_2074778), 1:4000 goat anti-mouse IgG-HRP (Santa Cruz Biotechnology, #sc-2005, RRID:AB_631736) and 1:4000 goat anti-rat IgG-HRP (Santa Cruz Biotechnology, #sc-2032, RRID:AB_631755). For chase experiments, 100 μg/mL cycloheximide (Sigma, #C7698) was directly added to the cultures. Detection was performed using SuperSignal West Pico Chemiluminescent Substrate (ThermoFischer Scientific, #34080), ECL Select reagent (GE Healthcare, #RPN2235) and a Fujifilm LAS-4000 imager. The ImageQuant TL software (1D gel analysis) was used for signal quantification. Measurements were statistically compared using two-tailed t-tests with the following p-value cut-offs for significance: 5E-2>*>1E-2; 1E-2>**>1E-5; ***<1E-5.

## RNA extraction and RT-qPCR analyses

RNAs were prepared using the hot acid phenol method and treated with DNaseI (New England Biolabs, #M0303S). 4 μg RNAs were used in reverse transcription reactions with 200 units of M-MLV RT (Invitrogen, #28025–013) and strand-specific primers. Following cDNA synthesis at 37°C for 50 min, the enzyme was inactivated at 80°C for 10 min. Samples were analyzed by qPCR with SYBR Green using a LightCycler LC480 apparatus (Roche) and quantification was performed using the ΔΔCt method. Controls without reverse transcriptase were systematically run in parallel to estimate the contribution of contaminating DNA. Amplification efficiencies were measured for each primer pairs in every run. Measurements were statistically compared using two-tailed t-tests with the following p-value cut-offs for significance: 5E-2>*>1E-2; 1E-2>**>1E-5; ***<1E-5. Oligonucleotides used in qPCR reactions are listed in **Supplementary file 2**.

## RNA-immunoprecipitation

50 ODs of cells were grown at 30°C in EMM 0.5X and harvested by centrifugation. Cell pellets were resuspended in 2 ml lysis buffer (6 mM $Na_2HPO_4$, 4 mM $NaH_2PO_4$, 150 mM $NaC_2H_3O_2$, 5 mM $MgC_2H_3O_2$, 0.25% NP-40, 2 mM EDTA, 1 mM EGTA, 5% glycerol, 1 mM AEBSF, 4 mM benzamidine, 2X Roche complete EDTA-free protease inhibitor cocktail and 160 U Murine RNase inhibitor (New England Biolabs, #M0314L)) to make 'pop-corn'. Lysis was performed using a Ball Mill (Retsch,

MM400) for 15 min at a 10 Hz frequency. Extracts were cleared by centrifugation before precipitation with 40 µL pre-washed anti-FLAG M2 affinity gel (Sigma, #A2220) for 2 hr at 4°C. Beads were then washed twice with IPP150 (10 mM Tris pH8, 150 mM NaCl, 0.1% NP-40). Total and immunoprecipitated RNAs were extracted with phenol:chloroform 5:1 pH4.7 (Sigma, #P1944) and precipitated with ethanol. RNA samples were treated with DNase (Ambion, #AM1906) prior to RT-qPCR analyses as mentioned above. Measurements were statistically compared using two-tailed t-tests with the following p-value cut-offs for significance: $5E-2 > * > 1E-2$; $1E-2 > ** > 1E-5$; $*** < 1E-5$.

Transcriptome analyses by RNA-sequencing cDNA libraries were generated according to standard Illumina protocols. The RNA sequences reported in this paper have been deposited in the NCBI Gene Expression Omnibus with the accession number GSE72327. Reads were trimmed with cutadapt and mapped to the ASM294v2.23 *S.pombe* genome using bowtie2. Read counts for every annotated transcript were calculated using HT-seq count and the ASM294v2.23 genome annotation. Differential expression was computed using the R bioconductor package DESeq2. Functional analysis was performed using the DAVID online tool.

### Ubiquitin pulldown

50 ODs of cells grown at 30°C in EMM 0.5X lacking L-leucine were harvested after 1 hr in the presence of 10 mM N-ethylmaleimide (Sigma, #E3876). Cell pellets were lysed with the NaOH/TCA method. Following centrifugation, pellets were washed twice with ice-cold acetone and then resuspended in 1.5 mL buffer A (6 M guanidinium chloride, 100 mM $NaH_2PO_4$, 125 mM Tris-base, 0.05% Tween-20), and incubated for up to 2 hr at room temperature on a nutator. Cell debris were removed by centrifugation and extracts were supplemented with 10 mM imidazole. 40 µL of pre-washed His tag isolation and pulldown Dynabeads (Life Technology, #10103D) were added and samples were incubated for up to 3 hr at room temperature on a nutator. Beads were then washed three times in buffer A containing 1 mM imidazole and four times in buffer C (8 M urea, 100 mM $NaH_2PO_4$, 100 mM Tris-base, 0.05% Tween-20, 1 mM imidazole) before elution in 30 µL sample loading buffer at 95°C for 10 min. For input samples, a fraction of extracts was precipitated with TCA, washed with acetone, air-dried and denatured in sample loading buffer at 37°C for 15 min. Samples were analyzed by western blotting using 4–12% NuPAGE gels, 1:3000 anti-HA antibody (12CA5) (Sigma, #11583816001, RRID:AB_514505), 1:3000 anti-FLAG antibody (Sigma, #F3165, RRID:AB_259529), 1:1000 anti-ubiquitin HRP-conjugated antibody (Santa Cruz Biotechnology, #sc-8017, RRID:AB_628423) and 1:4000 goat anti-mouse IgG-HRP (Santa Cruz Biotechnology, #sc-2005, RRID:AB_631736). Detection was performed using SuperSignal West Pico Chemiluminescent Substrate (ThermoFischer Scientific, #34080), ECL Select reagent (GE Healthcare, #RPN2235) and a Fujifilm LAS-4000 imager. Measurements were statistically compared using two-tailed t-tests with the following p-value cut-offs for significance: $5E-2 > * > 1E-2$; $1E-2 > ** > 1E-5$; $*** < 1E-5$.

### Microscopy

Exponentially growing cells cultured in minimal medium (EMM0.5X) were imaged at room temperature with a motorized Olympus BX-61 fluorescence microscope equipped with an Olympus PlanApo 100 × oil immersion objective (1.40 NA), Nomarski optics, a QiClick cooled monochrome camera (QImaging, Surrey, BC, Canada) and the MetaVue acquisition software (Molecular Devices; Sunnyvale, CA, USA). GFP-tagged proteins were visualized using a GFP filter set (41020 from Chroma Technology, Bellows Falls, VT; excitation HQ480/20x, dichroic Q505LP, emission HQ535/50 m). Images were processed in ImageJ (NIH).

## Acknowledgements

We are grateful to Sigurd Braun, Marc Bühler, Tim Humphrey, Hiten Madhani, Danesh Moazed and André Verdel for generous gift of strains and plasmids. We thank Antonin Morillon, Benoit Palancade and Tommaso Villa for critical reading of the manuscript. This work has benefited from the facilities and expertise of the high-throughput sequencing platform of IMAGIF (Centre de Recherche de Gif – www.imagif.cnrs.fr) and the structural and functional proteomic/mass spectrometry facility from the Institut Jacques Monod. FS was supported by PhD fellowships from the French Ministry of Research (Université Paris-Saclay) and the Fondation ARC pour la recherche sur le cancer (DOC20160603784). TC was supported by a PhD fellowship from the French Ministry of Research

(Université Paris-Saclay). This work was supported by the Fondation Bettencourt-Schueller (prix Coup d'Elan 2009) to DL and by the Fondation ARC pour la recherche sur le cancer (Projet Fondation ARC 1782) and the Agence Nationale de la Recherche (ANR-16-CE12-0031-01) to MR.

## Additional information

### Funding

| Funder | Grant reference number | Author |
|--------|------------------------|--------|
| Fondation Bettencourt Schueller | prix Coup d'Elan 2009 | Domenico Libri |
| Fondation ARC pour la Recherche sur le Cancer | Projet Fondation ARC 1782 | Mathieu Rougemaille |
| Agence Nationale de la Recherche | ANR-2016-CE12-0031-01 | Mathieu Rougemaille |

The funders had no role in study design, data collection and interpretation, or the decision to submit the work for publication.

### Author contributions

Fabrizio Simonetti, Tito Candelli, Sebastien Leon, Formal analysis, Investigation, Methodology, Writing—review and editing; Domenico Libri, Conceptualization, Resources, Funding acquisition, Validation, Writing—original draft, Writing—review and editing; Mathieu Rougemaille, Conceptualization, Resources, Formal analysis, Supervision, Funding acquisition, Validation, Investigation, Methodology, Writing—original draft, Writing—review and editing

### Author ORCIDs

Sebastien Leon http://orcid.org/0000-0002-2536-8595
Mathieu Rougemaille http://orcid.org/0000-0002-9675-3888

### Decision letter and Author response
Decision letter https://doi.org/10.7554/eLife.28046.023
Author response https://doi.org/10.7554/eLife.28046.024

## Additional files

### Supplementary files
• Supplementary file 1. *S. pombe* strains used in this study.
DOI: https://doi.org/10.7554/eLife.28046.018
• Supplementary file 2. Oligonucleotides used in this study.
DOI: https://doi.org/10.7554/eLife.28046.019
• Transparent reporting form
DOI: https://doi.org/10.7554/eLife.28046.020

### Major datasets
The following dataset was generated:

| Author(s) | Year | Dataset title | Dataset URL | Database, license, and accessibility information |
|-----------|------|---------------|-------------|--------------------------------------------------|
| Simonetti F, Candelli T, Leon S, Libri D, Rougemaille M | 2017 | Ubiquitination-dependent control of sexual differentiation in fission yeast | https://www.ncbi.nlm.nih.gov/geo/query/acc.cgi?acc=GSE72327 | Publicly available at NCBI Gene Expression Omnibus (accession no: GSE72327) |

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
