## [Decision Letter]

Thank you for submitting your article "Ubiquitination-dependent control of sexual differentiation by the Ccr4-Not complex in fission yeast" for consideration by *eLife*. Your article has been favorably evaluated by Kevin Struhl (Senior Editor) and three reviewers, one of whom is a member of our Board of Reviewing Editors. The reviewers have opted to remain anonymous.

The reviewers have discussed the reviews with one another and the Reviewing Editor has drafted this decision to help you prepare a revised submission.

The authors explore a role for Mot2, a RING domain containing subunit of the Ccr4-Not complex, in the regulation of meiotic transcripts. It was previously determined by other studies that Mmi1 associates with the Ccr4-Not complex. Mmi1 is an RNA binding protein that targets Ccr4-Not to meiotic transcripts for their repression. The authors confirm this interaction and set out to determine how Ccr4-Not regulates the repressed transcripts. Contrary to expectations, deleting the deadenylase subunits, Ccr4 and Caf1, does not affect significantly the levels of meiotic transcripts. The lone catalytic subunit of the complex that does is Mot2 (Not4). This leads the authors to explore a role of the reported ubiquitin ligase promoting activity of Mot2 in the regulation of Mei2, another RNA binding protein. They find that the amount of Mei2 protein is increased in a *mot2* mutant, which is dependent on the RING domain, suggesting its ability to recruit E2s is important.

The discovery that repression is not mediated by deadenylation and may require the ubiquitylation function is interesting. The experiments presented are mostly convincing and done well. A weakness is that it is questionable if Mot2 directly ubiquitylates Mei2. This is outlined below.

- No direct evidence that Mot2 affects the stability of the Mei2 protein, such as pulse chase, is provided. In fact, the attempt to measure half-life using CHX in Figure 6—figure supplement 1 suggests that the half-life is equivalent in the wild type and *mot2* mutant. There is also no direct evidence that Mot2 ubiquitylates Mei2. The in vivo genetic evidence contained in Figure 6 suggests the shifted band (presumably ubiquitylated Mei2) migrating between 130-250 is the same in both the WT and *mot2* mutant. The normalization to total protein is not convincing. Furthermore, the result that the overall levels of ubiquitylation detected by 6HIS pulldown in the figure is not surprising since Not4 has multiple substrates. Could the authors perform a reciprocal experiment of IP'ing Mei2 (in the presence of proteasome inhibitors) and measuring ubiquitylation levels of this protein?

[Editors' note: further revisions were requested prior to acceptance, as described below.]

Thank you for resubmitting your work entitled "Ubiquitination-dependent control of sexual differentiation in fission yeast" for further consideration at *eLife*. Your revised article has been favorably evaluated by Kevin Struhl (Senior Editor) and a Reviewing Editor.

The manuscript has been improved but there is one remaining issue that needs to be addressed before acceptance, as outlined below:

The genetics supporting Mot2's function in regulating Mmi1 is compelling. A significant gap, however, is the lack of evidence showing Mot2 ubiquitylates the suggested target. The effect on target turnover is also not solid. Since there is a limit to what can be done in the utilized system, the authors should, before acceptance, discuss these limitations of the study and that it is unclear that Mot2 directly ubiquitylates the substrate.

---

## [Author Response]

[…] The discovery that repression is not mediated by deadenylation and may require the ubiquitylation function is interesting. The experiments presented are mostly convincing and done well. A weakness is that it is questionable if Mot2 directly ubiquitylates Mei2. This is outlined below.- No direct evidence that Mot2 affects the stability of the Mei2 protein, such as pulse chase, is provided. In fact, the attempt to measure half-life using CHX in Figure 6—figure supplement 1 suggests that the half-life is equivalent in the wild type and mot2 mutant.

This is indeed an important point that we have seriously considered. We have shown that the impact of Ubr1 on the levels of Mei2 is much higher than the impact of Mot2. Based on the steady state levels of Mei2 in *ubr1∆* and *mot2∆* cells, it can be estimated that the rate constant for degradation of Mei2 by Ubr1 is roughly six times higher than the rate constant for Mot2-dependent degradation. This translates to only a difference of roughly 15% when comparing the half-life of Mei2 in wild type and *mot2∆* cells (e.g. 1.5 minutes difference for 10 minutes t_1/2_). Reliably measuring such a small difference is very arduous (if at all possible) considering the low levels of Mei2 in wild type cells and the constraints imposed by experimental errors.

We have attempted to improve detection of Mei2 in the wild type strain by using ^35^S-methionine for a chase experiment. However, and contrary to *S. cerevisiae*, we observed that ^35^S-methionine uptake was extremely inefficient in *S. pombe* (Author response image 1), which ultimately prevented use of this technique for the pulse chase. We also considered the possible existence of a very fast phase of Mot2-dependent degradation and attempted chase experiments with short time points after cycloheximide addition, but the general results were similar.

**Author response image 1. respfig1:** Kinetics of ^35^S-methionine uptake in budding and fission yeasts. *S. cerevisiae* (BY4742) and *S.pombe* wild type cells were grown to exponential phase (OD = 0.3 and 0.45) in YNB (Yeast Nitrogen Base, 2% glucose) and EMMO.5X media, respectively. 50 µCi ^35^S-methionine (EasyTag L-[^35^S]-methionine, NEG709A005MC, PerkinElmer) were added to 5 mL cultures and cells were collected using 0.2 µm filters at the following time points: 30 sec, 10 min, 20 min, 40 min and 60 min. Filters and aliquots of media depleted for cells were counted using a liquid scintillation counter (WALLAC 1409 DSA, Perkin Elmer) to determine the % of ^35^S-methionine uptake (filled lines) and the % of ^35^S-methionine remaining in the culture media (dashed lines). Blue and red curves refer to *S. cerevisiae* and *S. pombe* respectively.

Because of these experimental limitations, we have provided evidence that all the other steps in Mei2 production are not affected upon deletion of *mot2*+, notably transcription of the *mei2*+ gene, mRNA stability or mRNA translation (Figure 3 and Figure 5). Since the production of Mei2 is not positively affected in the absence of Mot2, the higher steady state levels of the protein can only be due to its slower turnover. Further supporting this notion, we have also shown that Mot2 and the proteasome function in the same degradation pathway (Figure 5).

In the revised version of the manuscript, we have stressed that degradation of Mei2 might not be the primary cause of its inactivation by Mot2. Indeed, it is possible that the primary effect of ubiquitination is to functionally inactivate Mei2 or to regulate its localization, and that degradation is a secondary consequence of Mot2-dependent ubiquitination.

There is also no direct evidence that Mot2 ubiquitylates Mei2. The in vivo genetic evidence contained in Figure 6 suggests the shifted band (presumably ubiquitylated Mei2) migrating between 130-250 is the same in both the WT and mot2 mutant. The normalization to total protein is not convincing.

Figure 6 indeed indicate that the amount of ubiquitinated Mei2 is similar in both wild type and *mot2∆* cells, but the total levels of Mei2 are significantly higher in the mutant. The only possibility to estimate the fraction of ubiquitinated Mei2 in each strain is to normalize to total Mei2 levels. This would not be correct if Mei2 were distributed in different pools, only some of which could be ubiquitinated, in which case normalization should be done relative to the fraction that can be ubiquitinated, but there is no evidence for this, neither in our experiments nor in the literature. For clarity, we now provide immunoblots showing the decrease of Ubi-Mei2 species in the absence of Mot2 when similar amounts of total Mei2 are probed from wild type and *mot2∆* cells (Figure 6—figure supplement 1).

To substantiate our claim, we analyzed the signals of Ubi-Mei2 in strains carrying the *mts2-1* proteasome mutation alone or in combination with the deletion of *mot2*+, which accumulate identical levels of total Mei2. In this context, we reproducibly observed a 2-fold decrease in the fraction of Ubi-Mei2 species in *mts2-1 mot2∆* versus *mts2-1* cells at the restrictive temperature (Figure 6—figure supplement 1), in agreement with a role for Mot2 in the ubiquitination of Mei2.

Another possibility is that the higher levels of Mei2 in *mot2∆* cells somehow saturate the ubiquitination pathway (e.g. Ubr1-dependent). To rule out this scenario, we compared the fraction of Ubi-Mei2 in cells expressing endogenous levels of the protein (e.g. Mei23xFLAG) or overexpressing it (e.g. P3nmt1-3xFLAG-Mei2). Even upon a roughly 100-fold Mei2 overexpression, we could observe a proportional increase in Ubi-Mei2 levels (Figure 6—figure supplement 1). This implies that ubiquitination is not limiting and that the decreased levels of Ubi-Mei2 in *mot2∆* cells are not due to an overproduction of Mei2.

These results, which strengthen our original conclusions, have been included in the revised version of the manuscript.

Furthermore, the result that the overall levels of ubiquitylation detected by 6HIS pulldown in the figure is not surprising since Not4 has multiple substrates.

We did not intend to claim that the overall levels of ubiquitination are decreased in the absence of Mot2. The lower signal of ubiquitin conjugates observed in the *mot2∆* mutant in Figure 6 was rather due to experimental variability. We now provide a new panel in Figure 6 in which the amount of these conjugates is similar between strains.

Could the authors perform a reciprocal experiment of IP'ing Mei2 (in the presence of proteasome inhibitors) and measuring ubiquitylation levels of this protein?

We attempted such an experiment, but we had to rely on the detection of species bearing endogenous ubiquitin. As suggested, we pulled down Mei2 from cells treated with the proteasome inhibitor MG132 but consistently failed to detect Ubi-Mei2 species from the wild type strain, most likely because the protein and/or endogenous ubiquitin are not sufficiently abundant, even in the presence of the proteasome inhibitor.

Overall, we have performed additional experiments and added new data to the manuscript, which largely support our model. Our results indicate that Mot2 ubiquitinates a low but substantial fraction of Mei2 to preserve the activity of Mmi1 in meiotic mRNA suppression. We would like to add that several lines of evidence further support a direct role for Mot2 in the control of Mei2 levels and/or function: i) Mmi1 physically brings Mot2 to Mei2, ii) Mei2 accumulates in mutants of Mmi1 that cannot associate with the Ccr4-Not complex, similar to *mot2∆* cells and, iii) the catalytic activity of Mot2 is required to limit the cellular amounts of Mei2. We hope that our experimental results will convince the referees and the editor that our model is correct.

[Editors' note: further revisions were requested prior to acceptance, as described below.]

The manuscript has been improved but there is one remaining issue that needs to be addressed before acceptance, as outlined below:'The genetics supporting Mot2's function in regulating Mmi1 is compelling. A significant gap, however, is the lack of evidence showing Mot2 ubiquitylates the suggested target. The effect on target turnover is also not solid. Since there is a limit to what can be done in the utilized system, the authors should, before acceptance, discuss these limitations of the study and that it is unclear that Mot2 directly ubiquitylates the substrate.'

Below are listed the modifications that we have made to the manuscript:

1) We have further discussed the limitations of our study by changing and/or adding the following sentences:

Discussion: “Our results indicate that Mmi1 recruits Mot2 to ubiquitinate its own inhibitor Mei2” was replaced by “Our results suggest that Mmi1 recruits Mot2 to ubiquitinate its own inhibitor Mei2, although we cannot formally exclude that the ubiquitination activity of Mot2 impacts the function or levels of Mei2 in an indirect manner.”

We also added “Because of this limitation, we cannot formally exclude the possibility that indirect effects may account for the increased Mei2 steady state levels in the absence of Mot2. Yet, we showed that higher Mei2 levels in *mot2∆* cells did not result from increased transcription and translation, pointing to the most parsimonious explanation that Mot2 impacts directly Mei2 turnover.”

2) We have modified the following sentences in the text:

“Biochemical and genetic analyses indicate” was replaced by “Our analyses suggest”.

“biochemical and genetic analyses indicate” was replaced by “biochemical and genetic analyses suggest“.

“but rather plays a regulatory role on a fraction of Mei2” was replaced by “but negatively controls Mei2“.

“these results indicate” was replaced by “these results are consistent with the notion”.

“in protein ubiquitination” was replaced by “in the regulation of protein ubiquitination”.

“However, our data exclude” was replaced by “However, our data argue against”.

“It is therefore” was replaced by “It is however”.

3) We have cited in the Introduction (first paragraph) a very recent work by Verdel and colleagues showing that Mmi1 targets additional mRNAs and lncRNAs for degradation by the exosome (Touat-Toschedini et al., 2017). The reference has been added.